# Cardiotocography in Obstetrics: New Solutions for “Routine” Technology

**DOI:** 10.3390/s22145126

**Published:** 2022-07-08

**Authors:** Vladimir Kodkin

**Affiliations:** Department of Electric Drive and Mechatronics, South Ural State University, 454080 Chelyabinsk, Russia; kodkina2@mail.ru

**Keywords:** cardiotocography, ultrasonic signal, heart rate (HR), beat-to-beat measurement principle, filtering, spectral analysis, simplified index of myocardial contractility, accurate calculation of heart rate

## Abstract

This work is devoted to the problems of one of the most common screening examinations used in medical practice: fetal cardiotocography (CTG). The technology of ultrasonic monitoring of fetal heart rate (HR) variations has been used for more than 70 years. During this time, it has undergone many upgrades and has been characterized several times as a hopelessly outdated routine technology. Over the past 5–7 years, many in-depth studies and review papers on cardiotocography have appeared, which revealed both the problems and prospects of the technology. Basically, hopes are associated with artificial intelligence, which should increase the accuracy of the analysis of initially inaccurate measurements obtained using ultrasonic testing. At the same time, after the introduction of pulsed operating modes and the appearance of multi-chip sensors, the quality of the original signal remains practically unchanged. This circumstance makes the prospects of the technology very problematic. However, until now, there has not been a reliable replacement for this screening, which is equally safe, non-invasive, and accessible to a wide range of specialists, medical institutions, and patients. The paper discusses and substantiates proposals for improving the technology based on original (different from traditional CTG) methods of processing information received from ultrasonic sensors, which, in the author’s opinion, allow for solving the main problems of CTG: identifying the correct direction of radiation to the fetal heart and to reliably evaluate beat-to-beat heart rate.

## 1. Introduction

In recent decades, cardiotocography (CTG) has remained the main method for screening the condition of the fetus in the prenatal period from the 20th week of pregnancy to delivery—a technology for diagnosing the intrauterine state of the fetus by variations in its heart rate (HR). This technology is used by diagnostic fetal monitors produced by many well-known companies. The most famous are Huntleigh Healthcare (England), Neoventa Medical AB—Sweden, Toitu—Japan, and others. The main advantages of CTG are non-invasiveness, safety, and relative cost-effectiveness. The method is based on the analysis of one-dimensional Doppler radiation reflected from the moving structures of the fetal heart.

Due to the Doppler effect, the frequency of the signal changes by a value proportional to the speed of movement of the structure that reflected it.

According to the frequency of the reflected signal, the devices calculate the speed of movement of the cardiac structures that reflect the beam, and according to the diagram of this speed, the periods of heart contractions are calculated as intervals between the peak values of the signal. By intervals, the frequency of contractions (HR) of the fetus and variations in heart rate over a time interval from 15 min to two or more hours are calculated. According to the calculated variations, the state of the fetus is analyzed.

It should be noted that over the past few years, many in-depth studies and reviews have appeared on the problems and prospects of cardiotocography. Thus, in the fundamental works [1,2,3], practically all known problems of the method are consecrated. For example, there are the absence of standards, the variability of the amplitude of the reflected signals, a mathematical formula that connects the Doppler effect with the speed of movement of the reflecting structure, the increment in the frequency of ultrasonic signals, and the angle between the radiation and the speed of movement. The review discusses in detail all the main methods for calculating fetal heart rate: from autocorrelation to spectral analysis. At the same time, all these methods are based on the detection of peak signal values received from ultrasonic sensors. These problems and prospects are also considered in a number of other works. However, even in the mentioned study, there is no analysis of the effect of the angle between the radiation and the reflective structure speed on the ultrasonic sensor output voltage, although the cosine of this angle is the same full-fledged factor in the voltage value, as well as the speed movement of the reflective structure. A lot of research is devoted to the sophistication and refinement of signal processing algorithms [4,5,6,7], and especially artificial intelligence technologies [8,9,10].

At the same time, the original processed signal remains uncertain and inaccurate. Algorithms for calculating fetal heart rate try to “highlight” the peaks of this signal, and autocorrelation algorithms choose the most reliable ones. As a result of these approaches, this necessary technology remains in the “screening” class.

Particular attention should be paid to the method of diagnosing the condition of the fetus by fetal ECG, recorded in a non-invasive way through the abdominal ECG of a pregnant woman, as is done by a number of devices, the most famous of which is the Monica monitor (England). This device is expensive and more difficult to maintain than conventional fetal monitors. It is very difficult to register a fetal ECG “at will”. One study using Monica will be mentioned below. Here we note the following. The prospects of this technology are obvious: it is the ECG registration that can be carried out continuously and provide the necessary information about the current state of the pregnant woman and the fetus. But this will happen, obviously, after a while. Another advantage of the technology is that if the fetal ECG is recorded, then its reliability is obvious and eliminates the need for probabilistic analysis. That is, cardiotocography registers a signal reflected from the structures of the fetal heart, almost constantly, but it is difficult to distinguish useful signals from false ones, and the fetal ECG is recorded less often, but the ECG signal is well distinguished from noise and interference. This paper proposes a method for processing ultrasonic signals, which will increase the reliability of the difference between signals reflected from large structures of the fetal heart, such as the ventricles and atria, from “false” signals reflected from other vessels and structures.

### Statement of the CTG Problem

The main problem of CTG is that the initial ultrasonic signal for calculations, reflected from the fetal heart, is unstable, non-stationary, and is not regulated by any documents. None of the devices show this signal for diagnostic purposes, only some of them show it as a clue to the correct direction of the ultrasonic beam, but devices often use sound for this purpose.

There are very few requirements for an ultrasonic signal: frequency and specific radiation power. There are no requirements for the shape of the signal, its spectral composition, and frequency characteristics of the processing equipment (unlike stringent requirements, for example, for electrocardiography (ECG)), there are no clear criteria for the correct direction to the fetal heart, as well as recommendations for which structures of the fetal heart it should be directed: atrium, ventricle or valves. Initially, the most important indicator in technology was the ability to keep ultrasonic radiation on the fetus, therefore, it was assumed that the heart rate is calculated from the movements of various structures and even large vessels. Apparently, this is due to the lack of regulation on the quality of signals and also the absence of regulations on the frequency characteristics of signal filters. At the same time, signal losses during diagnostics remained significant, especially in difficult cases and during childbirth. The conclusions made by various devices emphasized that “they are not a diagnosis.”

All this led to the fact that already 20 years ago the method was called “routine” and foreshadowed its imminent oblivion. But over the years, there was no replacement for him. Registration of the fetal ECG in a non-invasive way from the abdominal ECG of pregnant women, which had very high hopes, is still the field of research for a very limited number of devices (Monica, England), specialists, and medical institutions.

And examinations of pregnant women on complex ultrasound scanners are limited to 1–2 examinations per pregnancy for reasons of safety of pregnant women and economic feasibility.

## 2. Materials and Methods

### 2.1. About the USS Signal

It should be noted that in the works on CTG about the original signal, its features, and problems in the works devoted to CTG in recent years [1,2,3] almost nothing has been written. The waveform and its characteristics are accepted as predetermined and not subject to change [1,2,3]. Even in tutorials [11], one line limits the description of the sensors and how they are superimposed.

For a long time, a two-chip sensor was used. It contained one emitter and one receiver (Figure 1). In the last 30 years, sensors with one emitter and 5, 7, or more receiving crystals have become widely used. However, there are also few new research materials about two and multi-chip sensors. Very often, signal diagrams are given that are more reminiscent of two-crystal sensors and old algorithms for their processing [4,5,6]. As if multi-chip sensors did not give a qualitative change in the complex.

According to the mathematical description of the Doppler effect, given both in physics textbooks and in works on cardiotocography [1,12], ultrasonic radiation reflected from a surface moving at a speed *V* with an initial frequency *f* changes this frequency by a value proportional to the projection of the movement speed on the direction of radiation.
(1)∆f~V·cosφ

Here *φ* is the angle between the velocity *V* and ultrasonic radiation *U*. The signal obtained from the reflected signal by frequency detection:U = K∙V∙cosφ(2)

That is, by the value of this signal it is impossible to determine whether there has been a change in the speed of movement or the reflecting surface has changed the angle between its movement and the beam. At the same time, the spectral composition of these changes is quite close, and the chambers of the fetal heart change their position most likely with each heartbeat. Strictly speaking, it is impossible to distinguish between these movements by the signal isolated from the change in the frequency of the ultrasonic signal.

It is especially difficult to give an accurate estimate of the time of the peak value of the signal. Peaks (maxima) of the signal *U* and their temporal positions, in principle, do not coincide with the peaks of the cardiac structure *V* movement velocity.

To verify this, consider the signal and its time derivative. Let the angle between the speed of movement of the cardiac structure and ultrasonic radiation be a function of time *f(t)*, then:(3)U=V(t)·cosf(t); 

The extremum of the signal is determined by the zero of the time derivative of the signal. The derivative *U* is a rather complex function,
(4)$$\frac{dU}{dt}=\frac{dV}{dt}\cos f\left(t\right)+V\left(t\right)\left[-\sin f\left(t\right)\right]\cdot \frac{df}{dt}\;\;$$
and the times of its zeros may not coincide with the times of zero values of the derivative of the displacement speed *dV/dt*. It is impossible to estimate these differences from the signal.

It should be noted that the zeros of the original function *U* (Formula (1)) are determined only by the displacement velocity *V* zeros, or by the zeros of the cosine of the angle *φ* or the function *f*(*t*), that is, the perpendicularity of the displacement of the reflecting structure and ultrasonic radiation. The latter state is quite difficult to get stable. That is, the zeros of the *U* function in the signals of the fetal monitors are most often the zeros of the fetal cardiac structure movement speed.

Thus, it can be assumed that the accuracy of determining the reflecting structure velocity zeros is much better than the accuracy of determining the positions of its extrema.

### 2.2. About Ultrasonic Sensors (USS)

One of the main objects of research is the ultrasonic sensor. As already noted, about 30 years ago, almost all devices began to use multi-crystal sensors. There were no significant changes in signal processing. Judging by the latest works, almost all devices use half-wave signal rectification, that is, the signal for calculations reflects the absolute value of the structure speed of movement and does not respond to the direction of movement. In devices with dual-chip sensors, half-wave rectification was done in order to extract as many peak values of the signals as possible, no matter in which direction and from which structure the signal was reflected. The autocorrelation program played the most important role in the devices, which chose from all the measured intervals the most probable for calculating the fetal heart rate. This method did not make it possible to build a signal selection algorithm for accurate heart rate measurement on a beat-to-beat basis, which is considered necessary for a number of diagnostic techniques, and also created problems for signal certification, technology, and devices. A typical signal is shown in Figure 2.

At first glance, the signal contains a well-structured periodicity, however, if you count the number of extremums and the number of zeros on the diagram, the result will be discouraging.

The signal interval with a duration of 1.5 s, shown in Figure 2, which corresponds to 2–3 cardiocycles, contains more than 20 extrema, that is, the autocorrelation algorithm must reject 18 extremes. Figure 2 shows that one approximate cardiocycle will be characterized by 6–7 intervals measured between the local extrema of the *U*(*t*) function, from which it is necessary to choose only one most reliable, from which the next one will have to be counted with the same uncertainties. The accuracy of position determining of the extreme value of a signal of this form is better than 2–5 mS, even for high-precision measuring systems, is also in doubt.

The accuracy of almost any autocorrelation algorithm for the values of the intervals measured between the extrema of the processed function *U* will be very doubtful. It should be noted that the same function on the same time interval contains only 6 zero values, of which only 3 values need to be filtered.

The multi-chip sensor introduces one significant change in the logic of the technology. In Formula (1), instead of a multiplier, the sum of functions from different angles, spatially associated with one reflecting structure, appears. Such a measurement has much less signal loss, but the peak values of such a signal are an even more complex function and its extrema also differ from the extrema in the speed of movement. which is easy to see in the converted formula:(5)$$U=V\left(\sum^{\;}\cos\varphi_{i}\left(t\right)\right)\;\frac{dU}{dt}=\frac{dV}{dt}\cdot (\sum^{\;}\cos\varphi_{i}\left(t\right))+V\left(t\right)\left[\sum^{\;}\left(-\sin\varphi_{i}\right)\cdot \frac{d\varphi_{i}}{dt}\right]=0\;$$

In numerous works devoted to the improvement of cardiotocography, it is necessary to highlight such areas as autocorrelation of the measured time intervals, and the application of artificial intelligence. But all these methods extract the peak values of the reflected signals and estimate the probability of the reliability of the analyzed values. That is, the methods work in the absence of reliable signs of the measurement correctness. This does not allow, obviously, to formulate the principles of standardization of cardiotocography technology. One of the brightest examples of such technologies are swarm technologies.

### 2.3. SWARM Technology Method

In recent years, interest in the CTG method has suddenly grown stronger. Hopes are pinned on significant progress in software, in particular, on SWARM technologies and artificial intelligence, thanks to which useful signals are distinguished from a complex signal. The scheme of the algorithm is presented in [9] and in Figure 3.

Multiple processing of the original signal is applied, including half-wave rectification, filtering with flexible parameters, and the simultaneous use of several signals and a very large amount of information to select the most probable intervals.

For calculations, the signals of diagrams C and D are used, and with the help of artificial intelligence, the “correct” intervals between the peak values of the signal are selected. At the same time, signals with different spectra (C and D) are processed and analyzed. That is, almost the same signals are analyzed as in Figure 2. As well, if the error is inherent in the original signal itself, that is, the peak values of the signal of the C and D diagram do not coincide with the peaks of the reflecting structure movement speed, but are the peak values of the complex signal calculated by the Formula (3), the technology will not highlight these changes.

It should be noted that the SWARM filtering technology certainly has very good prospects in various scientific and technical fields and can be successfully applied in a wide variety of methods for extracting reliable signals.

### 2.4. New Methods for Which It Is Necessary to Measure the Heart Rate “from Beat to Beat”

In obstetrics, attempts continue to deepen prenatal screening diagnostics, in particular, the article [13] presents the results of the introduction into wide practice of the technology for detecting “deep bradycardia” to identify the risk of preterm birth.

In this case, it is required to analyze the exact values of the beat-to-beat fetal heart rate. The paper talks about a new diagnostic, it offers a theoretical contribution to the understanding of asymmetric fetal heart rate dynamics during term and preterm birth. In the article, they mentioned abdominal electrocardiography and CTG as the ways to collect initial information: intranatal heart rate monitoring using abdominal ECG can reduce “ambiguous fetal behavior”, traces of heart rate compared with cardiotocography, which is without outliers or missing data; incomplete clinical information and poor signal quality.

The use of Monica’s fetal ECG devices resulted in a “high-quality fetal ECG recording in only 17 out of 65 patients. With all the advantages of Monika’s devices and the proposed technology, it can be assumed that it is unlikely that this method will be applied in wide practice. Registration of abdominal ECG of pregnant women remains a problematic procedure for most obstetric institutions. So, is it possible to solve new problems of obstetric diagnostics with CTG devices?

## 3. Solution. Algorithm for Signal Processing in the «MAK02» Device

The improvement of algorithms for calculating fetal heart rate about 15 years ago in one of the Russian companies led to a somewhat different calculation method from the rest. Using a multi-chip sensor, the experts came to the conclusion that signal losses become quite rare and there is no need to catch all the reflected signals, ignoring their signs. They introduced a full-wave rectification of the reflected signal, that is, the circuit began to “feel” the direction of movement of the reflective surface.

The number of analyzed critical points of the signals immediately decreased, since it is sufficient to take into account the extreme values of the signals of the same sign to estimate the time of the period. Thus, the signal in Figure 4 is analogous to the Figure 2 signal, and requires the analysis of 6–8 extreme points for the entire interval and 3–4 for one cycle.

At the same time, the analyzed signal in the region of small signals quite often contained high-frequency chatter, which gave many false extreme and zero values. To eliminate this chatter, a low-pass filter was applied at the level of 6 Hz. This was significantly lower than the traditionally used filtering (35–40 Hz). But since there are no requirements for the filter, it is not necessary to “catch” the signals reflected from the valves, the filter has the right to exist.

In addition, the filter could not affect the signals reflected from large structures, since the multi-crystal sensor holds them well and there is no need to catch all the peaks of all signals reflected by the fetal vessels. Since the filter attenuated the signal, additional signal amplification was introduced. The effect was somewhat unexpected. Signals similar to those shown in Figure 4 took the form similar to the signals shown in Figure 5 by red diagrams. Arbitrary units of the processed ultrasonic signal are plotted along the ordinate axis, and time in seconds is plotted along the abscissa axis. Filtered and amplified signals reflected from large cardiac structures—the ventricle and atrium—became exceptionally recognizable, and the multi-chip sensor provided a fairly confident search for the desired state.

At the same time, a new problem appeared: the amplified signal often contained blurry peaks or even began to look like a trapezoid. With such signals, the exact time of the peak value cannot be found. The programmers found a way. Every second the signal spectrum was calculated and the main harmonic determined the heart rate in that second: a yellow graph in the diagrams shown in Figure 5, Figure 6, Figure 7, Figure 8, Figure 9, Figure 10, Figure 11, Figure 12, Figure 13, Figure 14, Figure 15, Figure 16, Figure 17 and Figure 18. The ordinate of this graph is conventional units, each of which corresponds to 10 beats per minute. (in all these figures, the yellow graphs are fetal heart rate charts calculated in the MAK serial device using the ultrasonic signal “extremum” calculation technology).

Within this second, the calculation is not specified. But all the nuances of variations—acceleration, deceleration, oscillation, etc., were captured by the calculation algorithm very accurately. Only the beat-to-beat calculation of heart rate has become, to put it mildly, conditional.

The main advantage of the algorithm is that the signal quite well identifies the direction to the heart, to a large structure, the ventricle or atrium, and if it hits another structure, the signal changes dramatically in the ratio of positive and negative sections and their shapes.

As follows from Formula (1), if the fetus starts to move, the frequency of the signal U increases sharply, and the diagram of the ultrasonic signal, shown in Figure 7 by the red line, confirms this. It differs sharply from the signals shown in Figure 5 and Figure 6. However, the “spectral” method for calculating the heart rate, which refines the calculation of the ultrasonic signal extrema in the “regular” algorithm of the MAK device: the yellow line in Figure 7, does not notice these “problems”. This confirms that spectral analysis is a method that is not capable of isolating rapid changes in the analyzed signals.

Let us return to the signals in diagrams 5 and 6. Long-term examinations of pregnant women and analysis of these data showed that such a signal, as shown in Figure 5 or Figure 6, when reflected not from the heart of the fetus, is impossible to get. That is, such a signal shape is the most reliable sign of work on the fetal heart, and there is no need for exclusively probabilistic analysis. Therefore, this algorithm solves the main problem of CTG. Such a signal and such processing can be proposed as a standard for examination.

Additional analysis of the signal showed that the problem of calculating the heart rate from the shock to the beat can be solved very reliably and confidently. Analysis of Formulas (1)–(3) showed that in the signal *U(t)* zero values are determined by the zeros of the velocity V of the structure of the heart or the perpendicularity of its velocity to the direction of ultrasonic radiation. It is obvious that the probability of a cardiac structure strict position is much less than zero speed.

It follows that the periods of heart contractions and heart rate according to the “beat-to-beat” principle should be measured by zero values of the rectified ultrasonic signal corresponding to a shape close to trapezoids [14,15,16,17].

It is possible to do this in a signal with a shape close to a trapezoid very accurately (Figure 8) [14,15]. The green charts in Figure 7 are the fetal heart rate, calculated “from beat to beat” over the intervals between zero values of the ultrasonic signal (red chart in Figure 7). Quite obviously that the green diagram reflects the variability of the reflected signal, in contrast to the yellow line.

Consider, for example, a 12-s CTG interval (Figure 9a,b), in which there were no dips or incorrect signals and the “regular heart rate calculation algorithm” (by spectra) showed minimal variations from 139 to 141 bpm (yellow middle diagram), and the calculation algorithm “by zeros” from 142 to 171 bpm (bottom green diagram) with a single false release up to 189 bpm.

In addition, it is a signal of this shape that allows you to accurately determine the phases of the fetal heart movement and highlight very clear large phases of cardiocycles—systole and diastole, which is extremely important for intranatal diagnostics [17,18].

To do this, it is necessary to measure the positive and negative signal intervals (T_1_ и T_2_ in Figure 8), presumably systole and diastole, and calculate a simplified version of the myocardial contractility index [14,15].

Consider the diagrams in Figure 9 and Figure 10. In each stroke, the intervals of movements of the cardiostructure in the forward and reverse directions were measured. The intervals vary: from 200 to 300 mS of the positive half-wave of the signal and from 100 to 200 mS of the negative half-wave of the ultrasonic signal. Thus, the simplified contractility index, calculated as the ratio of signal half-wavelengths, varied from 0.5 to 0.7, which is quite commensurate with the values determined by more complex and expensive instruments and methods [18]. Other well-known fetal monitors do not have such functionality. As mentioned above, the ability to keep the signal on the fetal heart has always been considered an important characteristic for the fetal monitor, that is, to maximize the time for reliable diagnosis and reduce the time for signal loss. It is also important to distinguish dangerous high fetal heart rate (tachycardia) from signals reflected from various moving structures and vessels. An analysis of numerous sessions showed that with tachycardia up to 180 beats/min, the signal retains a trapezoidal shape, and, if the fetus moves, then there is a clear failure of the signal, which is well distinguishable and will allow us to exclude this interval from the analysis. Moreover, the diagram of intervals, calculated from the zero values of the signal, very “clearly” indicates the “wrong” direction of the radiation.

At the same time, a number of new diagnostic possibilities appear: to determine the time of fetal activity automatically and to estimate pauses until a stable pulse is restored. In the diagrams (Figure 10, Figure 11 and Figure 12): pulses and a blurred form of the ultrasound signal immediately show the displacement of the heart from the direction of the beam. It should be noted that the “regular algorithm”: the yellow diagram does not capture this. Similar conclusions can be drawn from the diagram in Figure 13.

## 4. Results. Comments on Correct Diagrams

All diagrams given in the article are provided by the company that produces fetal monitors. The company is not listed in the pack because the article is not promotional material. The charts were obtained during planned voluntary examinations, according to the medical standard and requirements. The examinations were carried out in the interests of patients and were used to control their condition, the time of the examinations was more than fifteen years ago in several cities in Russia. The company used these charts earlier to refine their own heart rate calculation algorithms. Currently, the data are de-identified, and their demonstration in the article does not require the consent of the subjects.

In the diagram 14 (red line on the top), the processed signal, reflected from the heart of the fetus by ultrasound, with a deep 6 Hz filter amplified with “smeared peaks, but very clear zeros.” Zero “areas (isovolumic pauses) are not visible due to filtration, but negative ones are clearly defined and positive half-waves, whose duration ratios can be considered “screening” indices of fetal heart myocardial contractility (the ratio of the duration of systole to the duration of diastole).

The middle (orange) diagram is the fetal heart rate calculated by the device’s current algorithm based on the fundamental harmonic of the signal. Updated every second, but changes much less frequently than heart rate values calculated from zero values. The lowest green charts are fetal heart rate calculations based on signal nulls. Impulses show the separation between the negative wave of movement of the large chamber of the heart—systole, and the positive, diastole. The figure is the instantaneous heart rate for this signal.

Heart rate is ready for STV analysis or deep slowdown method for diagnosing preterm birth. In the diagrams of Figure 14, Figure 15, Figure 16, Figure 17, Figure 18, Figure 19 and Figure 20, there are also examples of the “correct direction” of ultrasound. The areas of movement of the reflective structure of the fetal heart, back and forth, are clearly visible. That is the compression of the chamber and its expansion in a first approximation, systole, and diastole.

The ratio of these intervals is estimated quite accurately. There are no signal peaks, but intervals and periods from beat to beat are very well calculated from zero values—the lower diagram and numerical beat-to-beat values of heart rate confirm this. In all the examples, the waveforms are very close to trapezoids, “recognizable” and well distinguished from the “blurred” signals in the diagrams.

## 5. Discussion

All the examples proposed in the work show that this signal processing from a multi-chip sensor, after full-wave rectification and filtering of signals above 6–10 Hz, reliably and reliably selects the signal reflected from large structures of the fetal heart. In addition, the proposed method for measuring time intervals “by zero values” of the ultrasonic signal allows you to accurately measure the fetal heart rate “from beat to beat” and with high accuracy not take into account false changes in the signal associated with fetal mobility. rather than changes in his heart rate. Standard devices that use the technology for measuring heart rate using peak values of an ultrasonic signal (like MAC) cannot do this.

This eliminates the need for probabilistic signal analysis when calculating the fetal heart rate and allows you to outline new effective diagnostic technologies that require reliable identification of cardiocomplexes and calculation of the fetal heart rate using the “beat-to-beat” method.

A well-known diagnostic technique for calculating fetal heart rate using this method is “short variations in fetal heart rate.” CTB is calculated accurately with a reflected signal like the one shown in Figure 19. One of the new technologies may be an approximate calculation of the ratio of positive and negative half-waves of the ultrasound signal, as a simplified index of contractility, which is necessary to determine the cardiac pathologies of the fetus in the prenatal state and is currently determined only in the process of examinations on ultrasound scanners, the number of which is recommended to be limited during pregnancy. It is possible to formulate some other proposals on diagnostic technologies for the analyzed ultrasonic signal, as in Figure 20.

Since cardiocycles are very recognizable in such signals, it can be proposed as one of the diagnostic characteristics: the number of heart beats over a long period of time: an hour, a day, as a measure of the work done by the heart, the “quantity” of cardiac activity. Previously, such characteristics could not be controlled due to too uncertain signals corresponding to cardiocycles. Another proposal is to make software and hardware that implements such an algorithm for calculating heart rate as a standard for all fetal monitors.

## 6. Conclusions

To calculate heart rate variations and diagnose the state of the fetus according to these variations, it is advisable to use the signals reflected from large cardiac structures, which most often have shapes close to a sequence of trapezoids, as the most reliable signals reflected from the fetal heart.

The studies have shown that only signals of this form should be used for diagnostic technologies and methods that require “beat-to-beat” heart rate calculations. That is, such a signal shape is the most reliable sign of work on the fetal heart, and there is no need for an exclusively probabilistic analysis.

Intervals with smeared signals, triangular in shape with a high rate of change or impulses, should be excluded from the analysis for diagnosing the condition of the fetus.

Calculations of cardio intervals should be carried out on the basis of zero values of the ultrasonic signal reflected from the fetal heart, processed according to the proposed method.

Using signals of this form, it is possible to conduct a screening analysis of the fetal heart myocardial contractility index in relation to the duration of positive and negative half-waves of the ultrasonic signal reflected from the fetal heart and conduct a prenatal assessment of possible fetal heart pathologies.

Frequency characteristics of filters with a bandwidth of 7–10 Hz and full-wave rectify cation of ultrasonic signals from multi-chip sensors should be proposed as a standard for CTG devices.

## Figures and Tables

**Figure 1 sensors-22-05126-f001:**
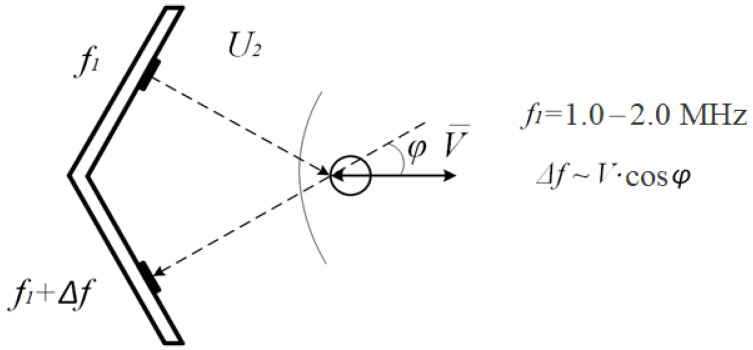
A sensor that uses the Doppler effect to detect the speed of movement V by changing the radiation frequency, f is the initial frequency of ultrasound, and f is the change in frequency by the Doppler effect.

**Figure 2 sensors-22-05126-f002:**
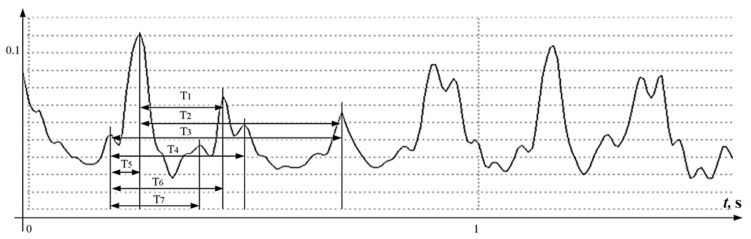
An example of a signal obtained after processing the reflected ultrasonic signal. T_1_–T_7_ Intervals between signal peaks.

**Figure 3 sensors-22-05126-f003:**
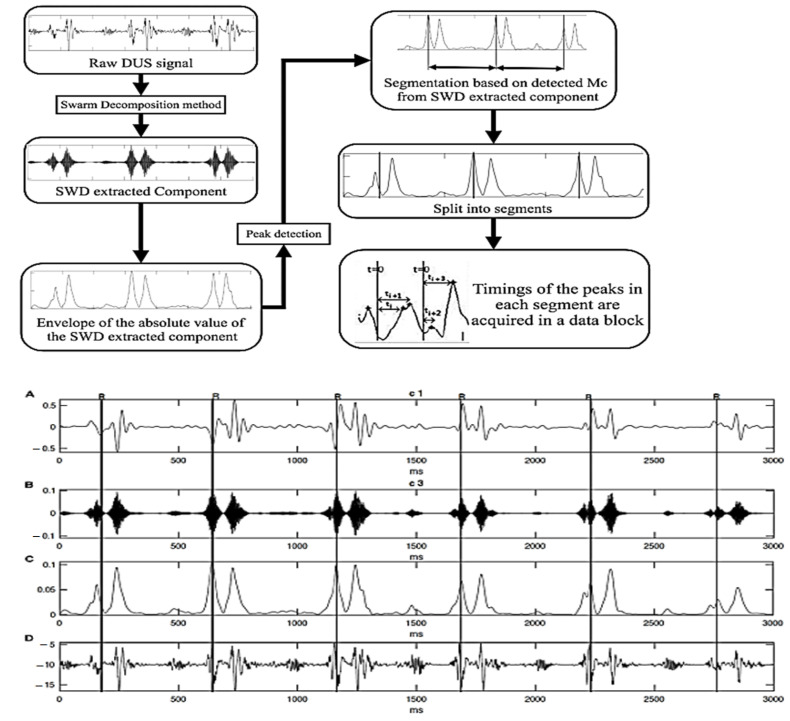
Schematic of the SWARM algorithm technologies [9].

**Figure 4 sensors-22-05126-f004:**
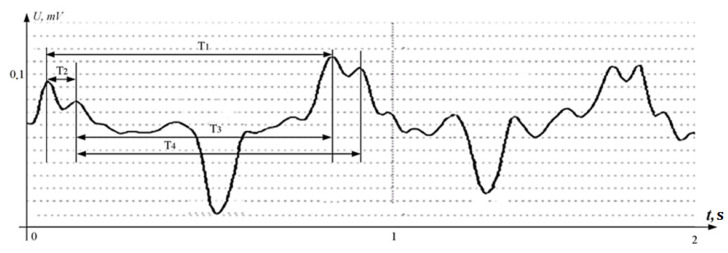
Ultrasonic signal reflected from the fetal heart after full-wave rectification.

**Figure 5 sensors-22-05126-f005:**
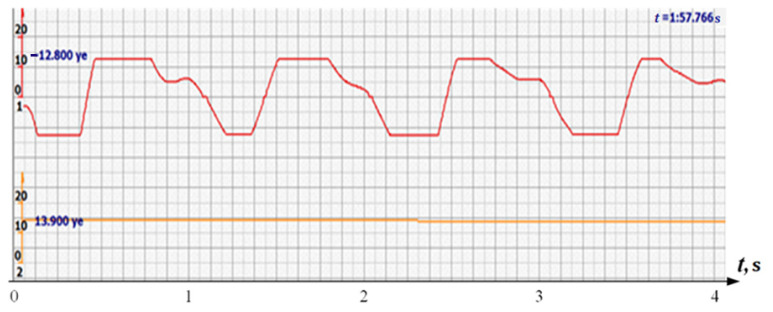
Ultrasonic signal (USS) conversion after filtering and amplification-red diagram. Heart rate determined by the fundamental harmonic of USS—yellow graph.

**Figure 6 sensors-22-05126-f006:**
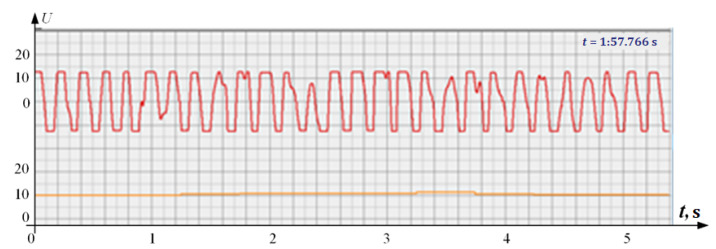
Diagram of heart rate calculations using the spectral method.

**Figure 7 sensors-22-05126-f007:**
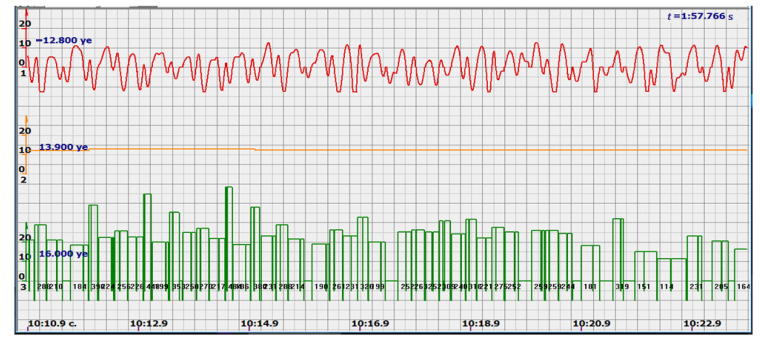
Diagram of an ultrasonic signal during fetal movement.

**Figure 8 sensors-22-05126-f008:**
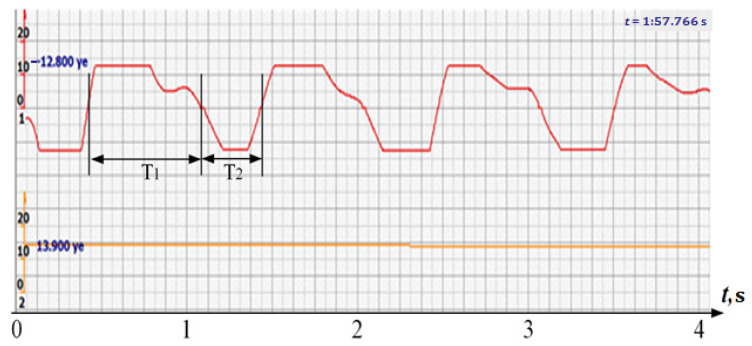
Measurement of the time intervals of the converted signal by its zero values—T_1_—”positive” half-wave and T_2_—”negative”.

**Figure 9 sensors-22-05126-f009:**
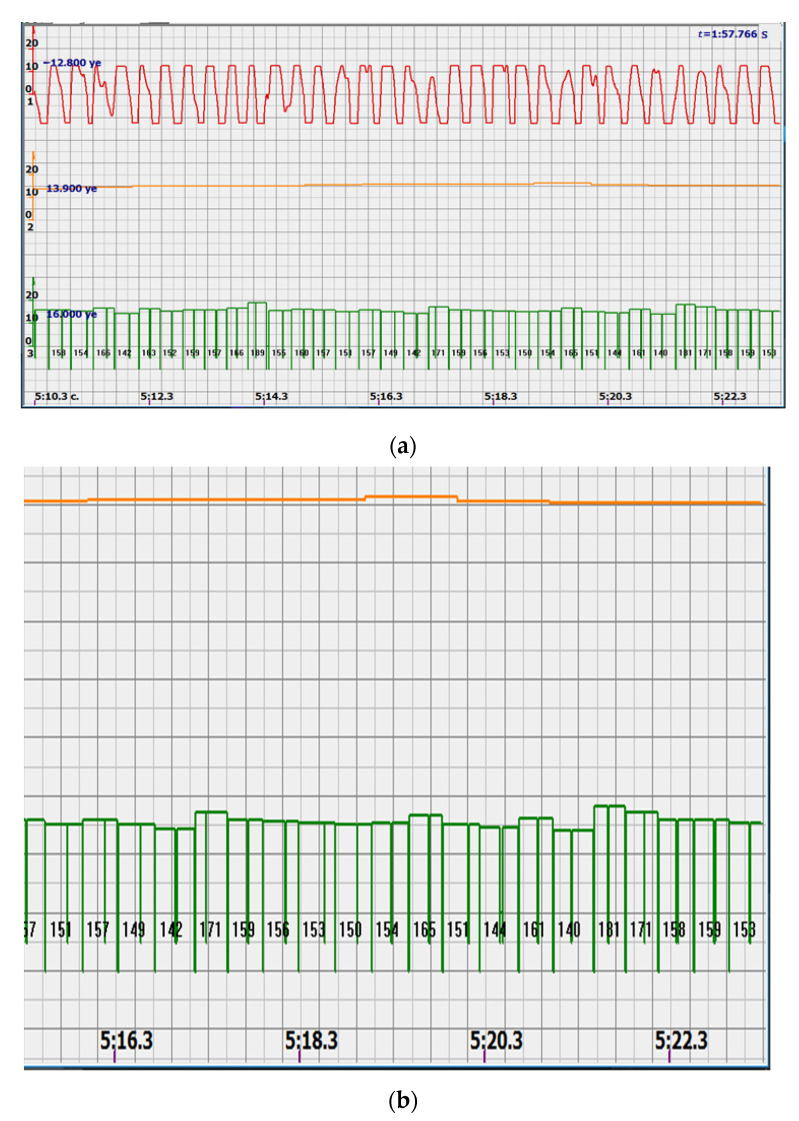
12 s interval of CTG (**a**), in which there were no dips or incorrect signals and the diagram of the fetal heart rate, calculated from the null values of the ultrasound signal (**b**), on an enhanced scale, along the ordinate axis, and in numbers: heart rate in beats in a minute.

**Figure 10 sensors-22-05126-f010:**
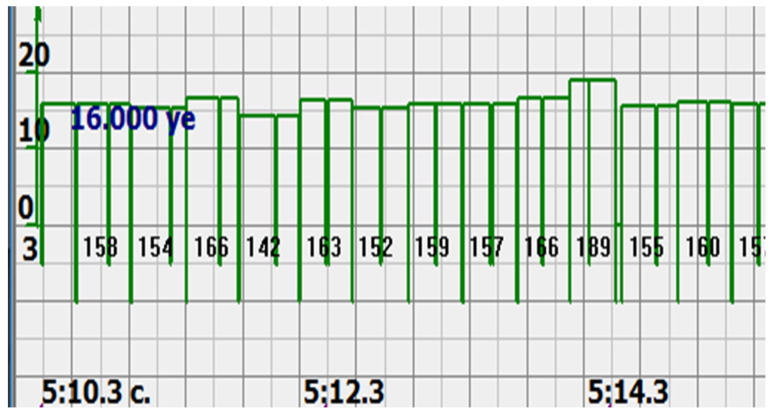
Heart rate intervals are divided by vertical lines into positive and negative half-waves in proportion to the duration of these half-waves.

**Figure 11 sensors-22-05126-f011:**
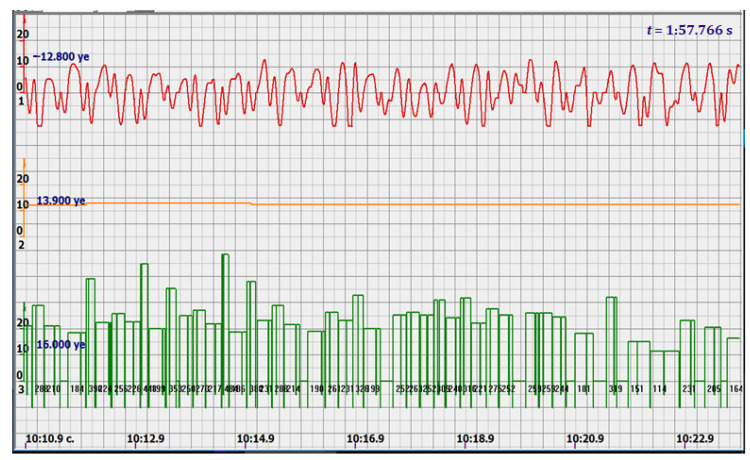
diagram during fetal movement—signal distortion and measured intervals.

**Figure 12 sensors-22-05126-f012:**
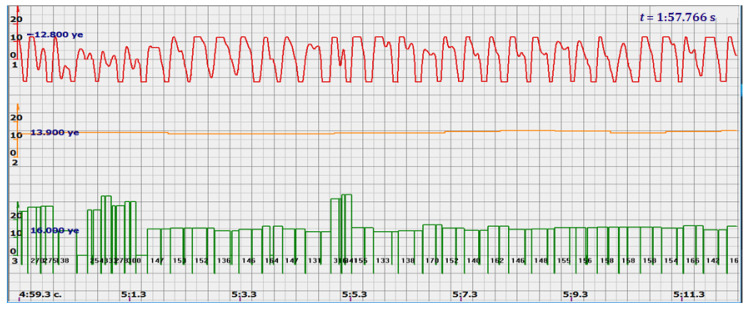
diagram during fetal movement.

**Figure 13 sensors-22-05126-f013:**
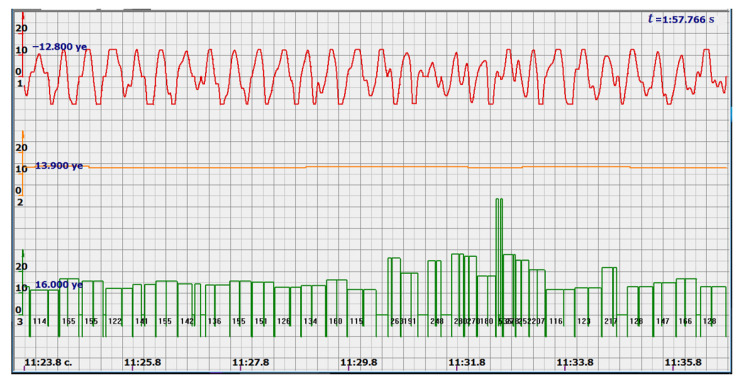
Fetal movements that are not responded to by the heart rate measurement system, which calculates the spectra of ultrasonic signals.

**Figure 14 sensors-22-05126-f014:**
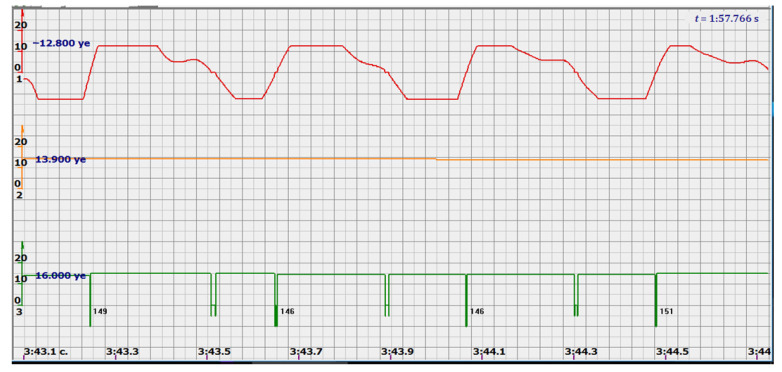
Fetal heart rate: from 146 to 151 bpm. Contractility index: from 0.5 to 0.65.

**Figure 15 sensors-22-05126-f015:**
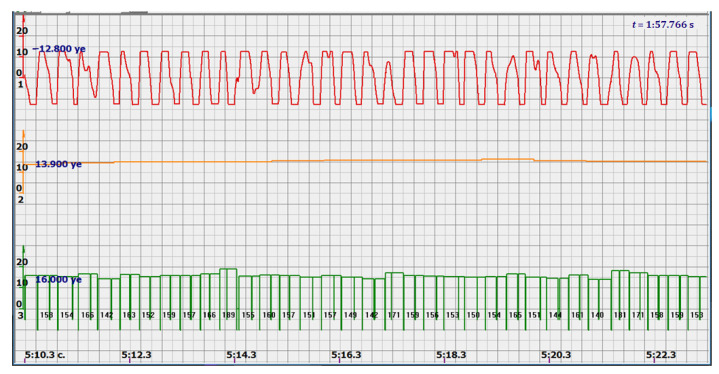
Heart rate—from 142 to 189 bpm.

**Figure 16 sensors-22-05126-f016:**
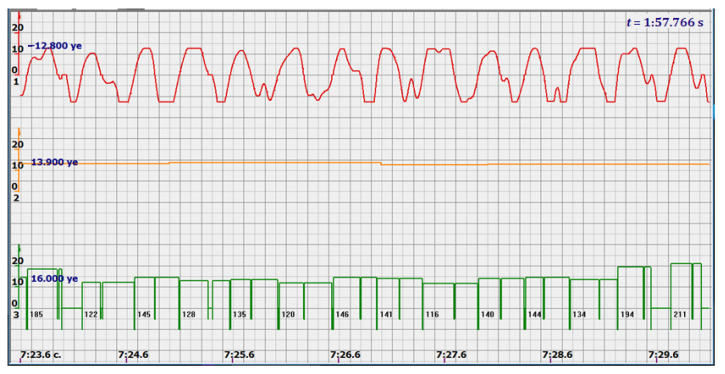
Heart rate—from 122 to 146 bpm.

**Figure 17 sensors-22-05126-f017:**
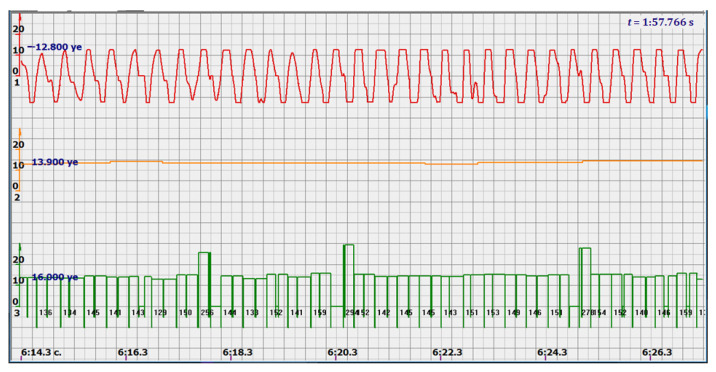
Heart rate—from 136 to 150 bpm.

**Figure 18 sensors-22-05126-f018:**
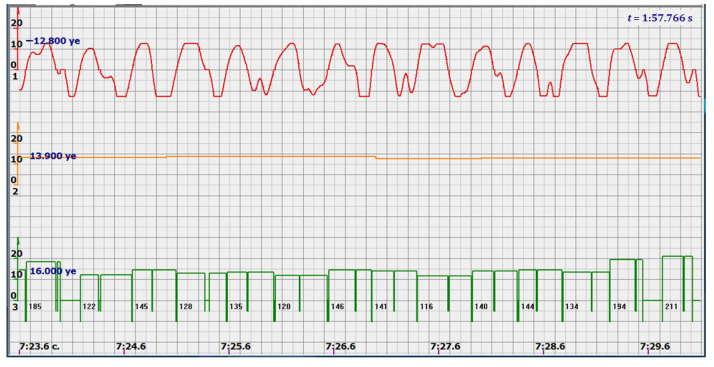
Heart rate—from 122 to 150 bpm.

**Figure 19 sensors-22-05126-f019:**
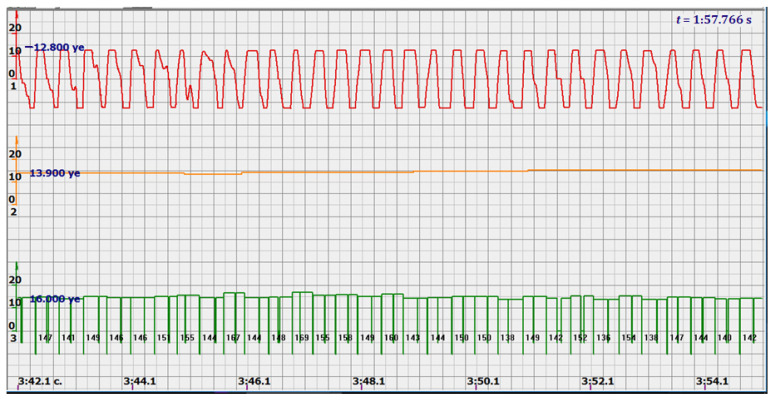
Heart rate: from 141 to 169 bpm.

**Figure 20 sensors-22-05126-f020:**
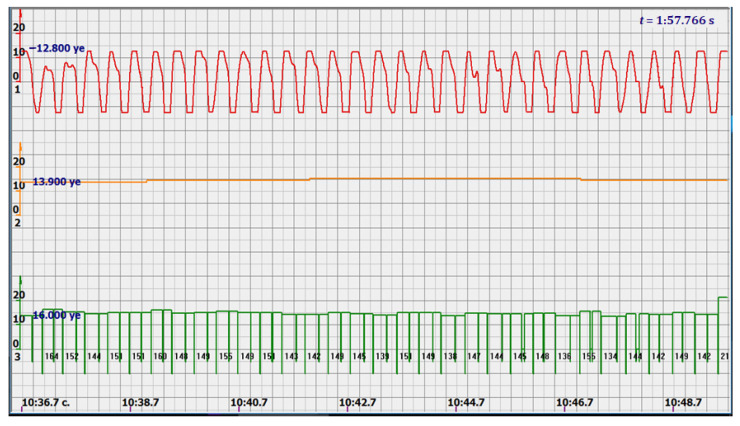
Heart rate: from 140 to 155 bpm.

## Data Availability

The raw data supporting the conclusions of this article will be made available by the corresponding author upon reasonable request.

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
