# Peer review of "Cardiotocography in Obstetrics: New Solutions for “Routine” Technology"

_sensors, 2022, doi:10.3390/s22145126_

Round 1
Reviewer 1 Report
The paper presents a research article titled as “Cardiotocography in obstetrics - new solutions for "routine" technology “. This paper presents very key studies and will prove very knowledgeable for its readership. However, incorporation of the following changes would further enhance the scope of this study:
- Literature in the introductory part seems in-adequate. Some more recent works should be included to update the relevant literature adequately. Additionally, there are no captions for Fig. such as Fig.1. Literature should be revised, and comments should be given for all figures.
- There are no numbers given for the equations. All the figures should be numbered, and equations should be revised.
- Figures 2, 3, 5 and 6 are having very poor quality which is not acceptable. Please improve the quality of the figures. These figures are mentioned and please have a look at others as well.
- Figure such as figures 11 and 12 are too messy and does not convey information.
- There are multiple grammatical and form errors observed. Authors should carefully go through all the manuscript and revise it.
- The introduction and conclusion should be revised. There should be no sub-heading in the introduction and there should be no numbering in the conclusion. The conclusion should be presented as a single paragraph.
- The abstract does not provide the significant information to understand the study.
- The challenges and limitations of previous studies overcome by this study should be listed in discussion section.
- The title of the paper seems too vague. It should be more specific.
- Keywords should be updated.
Author Response
answer in app

Reviewer 2 Report
The author researched a new way to detect a fetal heart rate with a non-invasive method. The methodology is based on the Doppler radiation reflected from the fetal heart by calculating the frequency of signal changes.
This is an exciting research report. In my opinion, the manuscript covers an important and current research topic. After addressing some comments and concerns, it should be suitable for publication in the Sensors Journal.
First, the author needs to reorganize the manuscript following MDPI's guidance for the authors. The manuscript seems to be rushed. For example, the hypothesis is presented before the conclusion. Also, an experimental setup needs to be clearly presented in one place. Further, there is no need to capitalize part of the sentence (L246).
There are no metrics in results discussion and comparison between obtained results and current research. Please, provide it.
Second, some comments and concerns found in the manuscript:
- Please, provide a short description for Figure 1.
- Equations in lines 85 and 88 are hard to read, probably due to pdf conversion. Further, the equations are not enumerated, at least not in my document. Please, enumarete them. Also, please, provide the source for the equations.
- Also, the equations in lines 102 and 105 are hard to interpret. Please, correct it, and provide the source for the equations.
- Please, provide a short description for Figure 2. Also, in the figure, time intervals from T1 to T7 are marked. Please, describe each one, and denote the y-axis.
- L147; From Figure 2, the six zeroes cannot be observed. Please, rectify it.
- Please, provide a short description for Figure 11.
Author Response
answer in app

Reviewer 3 Report
The paper was written non-professionally. Extensive editing is required.
All figures should be named. There are many short paragraphs. Some only have one sentence. This need to be rearranged and corrected.
It also needs better identification of how the proposed method benefit patients and what is the main disadvantages.
Author Response
answer in app

Round 2
Reviewer 1 Report
It should be accepted now.
Author Response
reply in app

Reviewer 2 Report
The author performed necessary changes and addressed comments and concerns to some degree.
First, academic writing has been improved but not enough.
Second, the main concern still remains. What is the main research contribution of the manuscript? Is it in algorithm L257 or a multi-chip sensor? It would be beneficial for the article clarity to draw a block schematic diagram of the proposed algorithm/method.
Third, how did you validate your results obtained from the proposed method? What was the referent device (obviously, you have used a MAK02 device at some point) to which you compared your obtained results?
There is a room for further improvements as folows:
- There are too many keywords, and they are too long, especially the last three added.
- In L155, please delete the word "formula." It is not necessary since equations are denoted with parentness.
- Paragraphs between lines 184 - 191 and 192 - 196 are almost the same. A noticeable difference is that in the first paragraph, Figure 2 is represented with 2-3 cardiocycles, and in the second paragraph, Figure 2 is represented with 3-4 cardiocycles. Please, correct it.
- Also, in L186, there is no need for three (3) exclamations as well as in L193. Please, use the discussion chapter to bring/emphasize the points.
- Two sentences in the paragraph that is contained between lines 197 - 199 should be explained in more detail since they are built up (the reason why this article is made) for your methodology.
- The sentence that begins with L240 contains the reference [12] two times cited.
- Figure 8 is not described.
Author Response
Reply in app

Reviewer 3 Report
Good written and presented. The format has been much improved. Can be accepted and published.
Author Response
Reply in app
